# Impact of the COVID-19 Pandemic on Palliative Care in Cancer Patients in Spain

**DOI:** 10.3390/ijerph182211992

**Published:** 2021-11-15

**Authors:** Cristina M. Beltran-Aroca, Rafael Ruiz-Montero, Antonio Llergo-Muñoz, Leticia Rubio, Eloy Girela-López

**Affiliations:** 1Section of Legal and Forensic Medicine, Facultad de Medicina y Enfermería, Universidad de Córdoba, 14004 Córdoba, Spain; cristinabeltran@uco.es (C.M.B.-A.); eloygirela@uco.es (E.G.-L.); 2Instituto Maimónides de Investigación Biomédica de Córdoba (IMIBIC), Hospital Universitario Reina Sofía, Universidad de Córdoba, Avenida Menéndez Pidal s/n, 14004 Córdoba, Spain; rafaelruizmontero@gmail.com; 3UGC Cuidados Paliativos, Hospital Universitario Reina Sofía, Avenida Menéndez Pidal s/n, 14004 Córdoba, Spain; antonio.llergo.sspa@juntadeandalucia.es; 4Department of Human Anatomy and Legal Medicine, Instituto de Investigación Biomédica de Málaga (IBIMA), Facultad de Medicina, Universidad de Málaga, 29071 Málaga, Spain

**Keywords:** palliative care, cancer, ethical issues, COVID-19, pandemic, legal medicine

## Abstract

Background: The COVID-19 pandemic outbreak has severely affected healthcare organizations worldwide, and the provision of palliative care (PC) to cancer patients has been no exception. The aim of this paper was to analyse the levels of health care provided by the Clinical Management Unit for PC in Córdoba (Spain) for cancer patients. Method: a retrospective cohort study was conducted. It analyzed the PC internal management database including all cancer patients treated in the period of 2018–2021. Results: 1967 cases were studied. There was a drop in cancer cases (*p* = 0.008), deaths at the PC hospital (*p* < 0.001), and referrals from primary care (*p* < 0.001). However, there was a rise in highly complex clinical situations (*p* = 0.020) and in ECOG performance status scores of 3–4 (*p* < 0.001). The pandemic was not shown to be a risk factor for survival in the PC program (0.99 [0.82–1.20]; *p* = 0.931). However, being female (*p* = 0.005), being older and having a high Karnofsky Performance Status (KPS) score (*p* < 0.001) could be indicators of a longer stay. Conclusion: The COVID-19 pandemic has presented a challenge in the management of patients requiring PC and has highlighted the urgent needs of the healthcare system if it is to continue providing a level of care which meets the needs of patients and their families.

## 1. Introduction

Medical care is not solely aimed at curing a patient’s disease, but also at alleviating illnesses [1]. Attending and caring for those who cannot be cured constitutes one of the main aims of medicine [2]. Since the mid-20th century, a branch of medicine has been developed which is specifically aimed at reducing the suffering of those patients whose pathology has advanced to a stage where they no longer respond to treatment, and where the possibilities of improving their chances of survival are limited [3]. According to the World Health Organization (WHO0, palliative care (PC) is “an approach that improves the quality of life of patients (adults and children) and their families who are facing problems associated with life-threatening illness”. Its goal is to prevent and alleviate physical, emotional, social or spiritual suffering associated with pain and other problems and to encourage the patients and their families to cope with suffering, death and agony with dignity [4].

The provision of “care” involves a series of exchanges and dynamic processes between those who provide it (the professionals) and those who receive it (the patient and their friends/relatives) [5]. Indeed, PC is deeply rooted in humanism, as it seeks to promote the moral qualities of recognising common suffering, fragility and humanity [6].

The WHO recognizes PC as a basic human right. Given these basic principles of prevention, wellness, science-based solutions and promotion, PC should be seen in a different light, and it should be considered a public health priority [7]. According to Powell, there are four types of humanitarian situations in which the provision of PC is particularly crucial: (i) long-term conflicts in which victims suffer from life-limiting diseases; (ii) major catastrophes in which victims are prioritized according to their chances of survival; (iii) outbreaks of contagious diseases with limited therapeutic options; and (iv) in refugee and displaced persons camps [8].

Worldwide, approximately 40,000,000 people, suffering from a wide variety of diseases, require PC. Under normal circumstances, as few as 14% receive the care they need [4]. If to this chilling statistic we add the incidence of the SARS-CoV-2 (COVID-19) pandemic in 2020, the full picture can be appreciated: this is a global crisis which has exposed the frailty of our healthcare systems [9].

The outbreak of the COVID-19 pandemic has profoundly affected healthcare organizations around the world [10], posing serious ethical challenges [6,11]. In particular, there have been many negative changes in the provision of PC services. The flow of patients and the delivery of services to provide complex, interdisciplinary, and person-centred care have been compromised by the fear of contagion by COVID-19 [12]. The situation has been particularly aggravated in certain conflictive social-political contexts in which a lack of equity in patient care was already evident [13].

The psychological consequences for the parties involved (healthcare professionals and patients) have not been long in coming. Even before the current pandemic, PC services have always registered higher mortality rates, which per se increase the risk of burnout among healthcare professionals, especially among nurses, due to their particularly close relationship with the patient [14]. Büntzel et al. state that, according to doctors, over 70% of cancer patients felt insecure, and up to 21% felt afraid and isolated [15]. They certainly had grounds for this fear, since the global prevalence of COVID-19 in these types of patients was found to be higher than in the general population [16]. In addition, patients believed that 34% of the medical staff were emotionally stressed or burned out [15]. In order to meet the demand for PC, the flexibility and adaptability of healthcare professionals and institutions has been essential. Although resources have been limited, the needs of the patients, their families and the staff have been met, and alternative solutions have been sought in the face of the rapidly changing requirements [17]. However, this situation has served to highlight the need to increase the available material and human resources and improve the conditions of access to PC services [18].

The aim of this work was to analyse a number of parameters related to the health care provided for patients at the Clinical Management Unit (CMU) for Palliative Care in Córdoba (Spain) during the COVID-19 pandemic for the years 2020–2021, and to note any possible differences compared with previous years.

## 2. Materials and Methods

### 2.1. Design

The study was a retrospective observational cohort study which analysed the parameters of healthcare provided to patients at the CMU for PC in Córdoba during the COVID-19 pandemic in 2020 and 2021 compared to the previous years.

For the purposes of the analysis, we established two different periods: the first was labelled “pre-pandemic”, running from 1 January 2018 to 14 March 2020, the date on which the ‘State of Alarm’ was declared in Spain, and including the lockdown which followed [19]. The second period was termed “pandemic” and ran from 15 March 2020 to 31 May 2021. No patient sampling was carried out: the records were taken from all those attended to at the CMU in the period under study.

The CMU consists of a PC unit providing inpatient care, plus three PC support teams for outpatient and home care.

### 2.2. Sample and Procedure

The sample was obtained from the CMU for PC internal management database and includes all the patients from central Córdoba who had been treated in the period from January 2018 to May 2021. All the cases selected were cancer patients, and a preliminary analysis was carried out to compare the incidence of other pathologies: cerebrovascular accident (CVA); chronic organic (cardiac, renal, hepatic, and respiratory) insufficiencies; neurodegenerative diseases: Parkinson’s disease, Huntington’s disease, dementias, amyotrophic lateral sclerosis (ALS); and other types of diseases, such as AIDS and other pathologies.

We excluded cases of patients under the age of 18 (*n* = 53), those who were referred on more than one occasion (*n* = 314) and those whose data showed a duration of the program or delay in admission which was negative or over 16 days (*n* = 32). Those patients who did not complete the follow-up from admission to the PC program until their death or the end of study (*n* = 13) or whose referral data were missing (*n* = 28), were also excluded. The final sample consisted of a total of 1967 cancer patients (Figure 1).

The parameters studied are listed in Table 1 and Table 2.

### 2.3. Data Analysis

All data were analysed using R version 4.0.3. Continuous measures were summarized as mean and standard deviation (SD) when the distribution of data was normal (Kolmogorov-Smirnov test); otherwise, as median and interquartile range (IQR). Categorical variables were reported as frequencies and proportions. In addition to the descriptive analysis, a comparison of proportions was made between the different groups by Chi-squared (χ^2^) test for contingency tables. Statistically significant differences for each category were obtained using corrected typified residues analysis (absolute values > 1.96). Wilcoxon rank-sum test was used to compare independent continuous samples (age and delay). A survival analysis was performed using Cox Proportional Hazard methods. The values considered to be statistically significant were those with a level of confidence of over 95% (*p* < 0.05).

### 2.4. Ethical Statements

All the data collected were anonymized and confidentially was guaranteed according to the Data Protection Regulation (EU) 2016/679 of the European Parliament and the Spanish Organic Law 3/2018. The study was conducted in accordance with the Declaration of Helsinki, and the protocol was approved by the Ethics Committee on Research in Córdoba, Spain (Project Ref. No. 5335).

## 3. Results

The total number of patients treated by the CMU for PC in Córdoba (Spain) in the period from January 2018 to May 2021 was 2834, of whom 69.4% (*n* = 1967) were cancer patients. When analysing the variations between the pre-pandemic and the pandemic periods (*p* = 0.008), no variations in the percentage of these patients were noted during the pandemic (pre-pandemic period vs. pandemic period: 70.2% vs. 68.1%). A slight decrease occurred with cerebrovascular accidents (CVA), while the incidence of neurodegenerative diseases increased (Figure 2).

### 3.1. Patients’ Socio-Demographic Data

59% of the patients were male, with statistically significant decrease observed during the pandemic, together with the corresponding increase in women (*p* = 0.027). There were no significant differences in the patients’ ages between the two periods. As regards the patients’ caregivers, the most common were first degree relatives (FDR) (85.3%). During the pandemic, a significant increase was noted (*p* = 0.008) compared to other relatives (nephews, etc.) or professionals (see Table 3).

### 3.2. Patients’ Clinical Situation

The most frequent type of cancer in the sample was lung cancer (20.7%), followed by colorectal (17.5%) and pancreas and bile duct cancers (10.1%). No statistically significant differences were observed in the results between the two periods. The existence of advanced stages of disease, with the presence of metastasis, was recorded in 39.8% of the patients, and no significant variations were noted. More frequently, the patients attended were not complex in nature (64.2%). However, during the pandemic, there was a significant increase in highly complex situations (*p* = 0.020) (Table 4).

When analysing the Karnofsky Scale (KPS) values (Table 2), 39.7% of the patients scored between 50–60 points. No variations were observed in the second period. On the ECOG scale (Table 2), patients who obtained a score of 2 represented 43% of all cancer patients and showed a significant decrease (*p* < 0.001) during the pandemic, with a corresponding increase in scores of 3 and 4 (Table 4). Finally, 98.5% of the patients attended to died, although during the pandemic there was a significant increase in discharges (*p* < 0.001) (Table 4).

### 3.3. Healthcare Provided

Up to 43.3% of the patients referred to PC were derived from outpatient care, followed by inpatient care (41.7%). The latter showed a significant increase in referrals during the pandemic, compared to the decrease in referrals from primary care (*p* < 0.001). In addition, over half of the patients (57%) were derived from the Oncology CMU, and 10.2% from Internal Medicine, a figure which did not vary between the two periods. In as many as 90% of cases, the patients’ referrals were not a high priority, with no significant differences between the two periods (Table 5).

The total time delay for the patient to be included in the PC program did not vary during the pandemic, with an average length of 1 day (0.00–0.00; *p* = 0.016). This result was due to the different distribution of the data (skewness and kurtosis values), so the *p* value has no clinical relevance. To estimate the length of stay of the patients in the PC program, we carried out survival analyses, as shown in the following subsection, corresponding to the variables which could act as conditioning factors.

Finally, in 53.2% cases (*n* = 1938), patients’ deaths occurred at home, while the palliative hospital (30.3%), showed a significant decrease in the second period (*p* < 0.001) (Table 5). This trend increased during the pandemic.

Table 6 shows how there was a significant decrease in score 2 on the ECOG scale and an increase in patients with score 3 regardless of whether they were referred from primary care, outpatient or inpatient care.

#### Risk and Survival Analysis for the Permanence of Patients in the Program

As can be seen in Table 7, the Hazard Ratio (HR) value indicated that the pandemic did not become a risk factor for the patients’ survival or permanence in the PC program (0.99; *p* = 0.931). The risk was reduced by 16% with a higher KPS score (*p* < 0.001) and by 26% in patients referred from outpatient care (*p* = 0.012), compared to primary care. Similarly, the oldest patients showed an 11% reduction in the risk of a shorter stay in the program, compared to the youngest (*p* < 0.001). Women also had a lower risk (21%) compared to men (*p* = 0.005). Being older and female could therefore lead to a longer stay in the PC program. The opposite effect was observed in the presence of metastasis (*p* < 0.001) and higher scores on the ECOG scale (*p* = 0.024).

### 3.4. Degree of Patients’ Knowledge and Preferences

1.1% of cancer patients had registered an advance directive living will (ADLW), with no variations during the pandemic.

54.1% of the patients were fully aware of their current clinical situation, while 12.9% had not been informed. Both percentages saw a non-significant decrease during the pandemic. On the other hand, more than half of the patients (59%) were able to fully assess the implications of their clinical condition, with a significant increase during the second period (*p* < 0.001). As regards family members/caregivers, the vast majority were also aware of the clinical aspects related to the patient’s situation (95.8%), while a very small percentage had not been informed (0.3%). Following the same trend, the majority (90.1%) were aware of the diagnosis and condition of the patient involved. In no cases were any statistical differences observed between the two periods (Table 8).

## 4. Discussion

During the pandemic, the main objective for health centres was, first and foremost, to contain the spread of the virus and care for patients infected by COVID-19, which had inevitable repercussions on patients with chronic non-communicable diseases, such as cancer [23]. It is a well-known fact that the pandemic has affected the number of cancers diagnosed in many countries, and the actual number of cancers diagnosed in 2020 was lower [24]. In contrast to our study, a slight decrease in the number of cancer patients attended by PC was generally observed, although the figures vary when the literature is reviewed [25,26]. AlShehery et al. reported a slight fall in the figures between March and June 2020 compared to 2019 (346 vs. 319). Meanwhile, although the lockdown period seems to have led to an initial decrease in the number of patients coming to hospital, the overall number of patients attended was not affected compared to previous years [27]. In one general cancer centre in France, for instance, a decrease in the number of patients was noted, especially during the lockdown period. However, this centre also registered a significant increase in hospital admissions for PC, after referrals from other hospitals, together with a fall in the number of diagnostic tests conducted and treatments applied (especially RT sessions, surgical procedures and admission to clinical trials) [28]. In contrast, there have also been instances in which no relevant changes have been reported, as is the case in our study [29]. What we did note was a change in the origin of referrals to PC. The impact of the pandemic on the dynamics of primary care caused a significant decrease in the number of patients referred and a lower survival rate in the PC program compared to those referred from outpatient care. Despite this, our data show that continuity of care in PC could be maintained. In this regard, the most evident result was that although it was not possible to quantify that telephone care was predominant during strict confinement, there was no delay in the admission of patients to the program.

### 4.1. Patients’ Socio-Demographic Data

On a worldwide scale, cancer has been recorded as the leading cause of death in men and the second leading cause in women [30]. In line with previous results [31], we observed that more than half the patients seen in the CMU for PC were men, although the number of women increased and that of men decreased during the pandemic. The literature reveals an uneven picture: some studies note no significant changes between the two genders [29], while in others, the percentage of women was higher before the pandemic [27].

The cancer patients analysed in our series showed a median age of over 70 years, which is much higher than in other research (50–70 years) [27]. Previous references have recorded an increase in younger patients in both outpatient and hospital care during the pandemic [29], while we did not note any changes between the two periods.

The pandemic also saw a slight fall in the number of cancer patients and therefore in the number of caregivers. Fear of contagion reduced the number of interactions by creating a restrictive environment [11]. Social distancing measures affected the availability not only of the health services, but also those of social support, placing an additional burden on caregivers, many of whom have cut themselves off from normal family and social networks in order to fulfil their obligations [32]. This situation could be responsible for the increase in the number of FDRs as main caregivers, while the number of more distant relatives and even salaried professionals as caregivers has fallen significantly.

### 4.2. Patients’ Clinical Situation

Cancer patients are considered more susceptible to severe COVID-19-related illness and death [33], mainly due to immunosuppression, as a consequence of the underlying neoplasia or of anticancer treatments. In particular, having received chemotherapy (in the last three months) or ongoing extensive radiotherapy, leukopenia, being over 60 years old, hospital admission and frequent hospital visits are risk factors which predispose cancer patients to catching the SARS-CoV-2 virus [34].

Protection policies in caregiving have been applied worldwide in order to lessen the exposure of these patients to this health risk [35].

Clinical trials and screening programs were affected, to a greater or lesser degree, and a degree of uncertainty was generated [24,36]. Penel et al. linked this reduction in detection to a decrease in the diagnosis of early-stage colorectal or non-palpable breast cancers [28]. However, since the present work concerned patients treated with PC, this information was not available in our case. In fact, the incidence of different types of cancers was similar before and during the pandemic, while in the specific cases of colorectal or breast cancer, the percentage of patients suffered a slight upward trend, although this was not significant.

As regards treatments, a range of curative and palliative therapies were at first delayed, shortened or modified, despite the lack of evidence to support taking this drastic step [37,38,39,40]. The safety of these treatments was later demonstrated [41], and patient prioritization resumed following strict protocols [42]. These changes were thought to have increased the likelihood of disease progression and the incidence of metastatic disease [43,44]. The results of our study show a not significant increase in the incidence of metastasis during the pandemic, although it was a risk factor for a shorter stay in the PC program. On the other hand, changes were observed in the categorization of the complexity of the patients seen, with a significant increase in highly complex situations during the pandemic. This assessment depends not only on the patient in terms of their clinical/psycho-emotional status, but also on the family/environment and the organization of health resources [5]. According to the diagnostic instrument (IDC-Pal) used to assess the complexity of the patient’s situation, (i) if the situation is not complex, the intervention of advanced/specific PC resources is not required; (ii) if it is complex (≥1 element of complexity), it is up to the responsible physician to decide whether to provide these resources; and (iii) if it is highly complex (≥1 element of high complexity), the intervention of advanced/specific PC resources will be required [20]. The significant increase in the care of patients with highly complex situations (19.3% vs. 26.8%) could be the outcome of all the above-mentioned changes made during the pandemic. These results are in line with an evident worsening in the quality of life of the patients attended, with more scores of 3 and 4 on the ECOG scale, added to a decrease in almost asymptomatic patients with scores of 2. With regard to the assessment of functional condition, the KPS did not show any significant variations. Scores of >50 fell and scores of <40 rose, which was logical considering the increased limitations in the patients’ ability to perform daily tasks and self-care. In addition, this tallies with the trend towards admitting cancer patients in more advanced stages during the pandemic and with the patients attended in PC having more limited chances of survival.

### 4.3. Healthcare Provided

In health centres and oncology services, there was a decrease in the rate of patients referred for a first diagnosis or treatment, mainly due to travel restrictions and the patients’ fear of contagion, leading to many patients not wanting to start systemic treatment or attend the clinic [45,46]. Büntzel et al. reported that up to 74% of oncologists took longer to convince patients to receive curative or palliative treatment due to their physical and emotional stress [15]. This reduction, beginning in primary care, was clearly seen in oncology units [45,47]. In addition, the cases studied in our series have enabled us to verify that PC was also significantly affected [27]. The difficulties in primary care stem from healthcare, organizational and ethical problems. Scheduled activities were cancelled, there was little home care or follow-up of chronic patients, protocols were changed, and patient care was almost exclusively provided by phone [48]. The lack of contact with the health services also produced a decrease in referrals from outpatient care, while those from inpatient care increased. These patients came mainly from the Oncology CMU, and no significant variations were observed between the two periods or between the different services.

Over the last decade, the benefits of receiving early PC have become increasingly apparent. For this reason, the American Society of Clinical Oncology (ASCO) recommends that patients with advanced cancer should be attended by PC teams within eight weeks of diagnosis [49,50]. The response of the CMU for PC teams follows a strict protocol according to the patient’s situation. Three levels of priority have been established: normal (maximum care period < 10 days), preferential (<5 days) and urgent (<48 h) [51]. In our study, no significant changes were observed in the patients’ levels of priority over the two time periods. In addition, we must bear in mind that this CMU has an established protocol of attending to patients on the same day, regardless of their level of priority. In some cases, this attention is provided by phone, and a visit is scheduled as soon as possible, depending on the circumstances. Not only is there no delay in attending to the patients, but they are also admitted to the PC program immediately. As a result, the maximum time allowed for a patient to be assessed and admitted is, on average, one day, according to the internal rules of the CMU. We can therefore confidently assert that no delays in patient care were detected during the pandemic.

Although the activity carried out by the PC unit is difficult to assess, its activities differed in the two periods, even during the pandemic. At first, during the months of March and April 2020, there were significantly fewer home visits, and there was a generalised ignorance about the disease, fear of contracting it from professionals, patients and relatives, and a systematic use of individual protection equipment in feverish patients [9]. All of this led to a significant fall in home visits and a rise in the number of telephone appointments. Normal service was resumed in July 2020.

Another of the parameters analysed was the length of the patients’ stay in the PC program. Chou et al. and AlShehery et al. studied the length of stay of patients admitted to PC units and observed a non-significant reduction of two days during the pandemic [27,29]. Unlike other authors [52], we were unable to confirm whether the variations caused by the pandemic (interruption of services, delay in treatment, type of care provided, etc.) constituted a risk factor which increased cancer patients’ mortality. When patients were diagnosed in outpatient care, their length of stay in the program was greater compared to those who came from primary care. The decrease in face-to-face appointments in primary care made it harder to maintain the continuity of care, thus reducing the opportunity to detect symptoms of recurrence early on [38]. These results are consistent with the fact that patients referred with preferential or urgent priority had a lower survival rate within the program.

As in previous studies, age and gender were considered determinants of patient survival. Older cancer patients usually have a worse prognosis [53], especially in the first months after diagnosis [54]. However, there is little evidence to show the direct and indirect impact of the pandemic on the survival of elderly patients [55,56]. Our results showed survival and shorter stays in PC in younger patients, which could have been due to their more severe clinical situation when admitted to the program. According to Pendyala et al., this evidence could reflect the fact that elderly patients took extra precautions through fear of contagion by COVID-19, which resulted in a higher survival rate. In this study, a positive correlation was shown between age and the frequency of changes in patient care during the pandemic [57]. On the other hand, another factor could be the socio-sanitary conditions resulting from the pandemic and the greater, more aggressive therapeutic treatment given to younger patients. As regards gender, women had a higher survival rate than men, possibly due to the fact that the most frequent cancers in the study (lung, colorectal and pancreas) accounted for 48.3% of the total and presented a worse prognosis in male patients [54].

One high priority at the end of life is to ensure that dying and death itself occur in the most comfortable way possible, and it is at this time when PCs are associated with a higher level of quality of care [58]. In the period 2015–2021, there were 1,044,856 deaths from causes of death alleviated by PC in Spain. Contrary to our results, most of these patients died in hospital, and where the cause was oncological, the probability was 75% higher compared to death at home [59]. Despite this, up to 80% of patients with advanced cancer choose their own home as the preferred place of death [60]. The quality of dying and death is considered higher when patients die at home, and it is positively correlated with the quality of care [61]. It is therefore a high priority to ensure the availability of PC services [62], especially home support teams [63]. In line with previous studies [64], a significant drop in deaths in palliative units was observed during the pandemic, while the number at home increased. This may have been due partly to family members’ fear of contracting the disease in the hospital. In addition, due to the strict limitations on accompanying dying patients, which were limited by the regulations to a single companion, it was often impossible for family members to give their loved ones their final farewells. These are precisely the issues which have led to a steady growth and development in PC services over the last fifteen years, a trend which has been noted in most European countries [65].

### 4.4. Degree of Patients’ Knowledge and Preferences

Advance planning of care has a positive impact on the quality of care at the end of life, with unwanted supportive treatments reduced, the use of PCs increased, and unnecessary hospital stays avoided [66,67]. These plans have been shown to be more effective in satisfying patient preferences than documents written by the patients themselves, such as an advance directive living will (ADLW) [67]. The present study was unable to assess the advance planning of care, but it did analyse the existence of one ADLW document, although the frequency of its use was negligible (around 1%), and no changes were observed during the pandemic.

Finally, data was collected about whether the patients and their families were aware of the patient’s current clinical situation, and whether or not they understood the outcome of this situation, in other words, death, in almost 99% of cases. Fortunately, most patients and relatives were fully informed of the patient’s position and understood the consequences of the diagnosis. Here, the numbers of fully-informed patients increased during the pandemic, which clearly reflects the good work carried out by the PC team. The early participation of the PC team also helps them to better understand the patient’s needs and to prioritize their values. Furthermore, if both patients and relatives are fully aware of the clinical reality and actively participate in decision-making throughout the process, greater benefits accrue for the patient and a lower risk of morbidity is achieved [68]. During the pandemic, bereavement has been a constant, widespread phenomenon affecting patients, families and healthcare professionals. The mourning process has been transformed by lockdown laws, restrictions on hospital visits, the uncertainty of a possible contagion and the difficulties of holding a funeral. Once again, the training and experience of the PC team have proved a crucial resource here in improving communication and support for patients and families, as well as in the planning of advance care and self-care [69].

There are a number of limitations when it comes to generalising these results. For instance, the sample we analysed was cross-sectional in character. In addition, since the study was limited to the CMU for PC in Córdoba, it is not possible to extrapolate the results to other health centres. Moreover, although the sample size was representative, a considerable number of cases with incomplete data had to be discarded.

## 5. Conclusions

Our results confirm previous findings, in that we found a slight decrease in the number of cancer patients treated by the CMU for PC during the pandemic. Among the possible explanations for this include the social distancing measures imposed to avoid contagion, which led to a change in the patients’ main caregivers, and a notable decrease in referrals from primary care. Another outcome observed was a lower number of deaths in the PC hospital compared with those in the patient’s home. Although the delays in the admission of patients to the program did not increase in length, the number of patients with more complex clinical situations and with lower scores on the ‘quality of life’ scales increased. The survival and length of stay in the PC unit of younger patients was lower, probably as a result of greater precautions being taken in older patients and the younger patients having a worse clinical situation on admission to the program.

On the other hand, our study was unable to demonstrate that the COVID-19 pandemic was a risk factor for patients requiring PC. However, it is clear that managing these patients has been a daunting challenge, and our health system has had to adopt urgent measures in its efforts to meet the needs of patients and their families.

The unprecedented effects of COVID-19 on healthcare systems have resulted in an unavoidable substantial impact on global health. Cancer patients in palliative care are not only dealing with the burden of their diagnosis and frailty, but also with the risk of being more susceptible to severe disease from COVID-19.

Our study highlights the need to maintain an adequate palliative care support network to provide optimal patient care. At the same time, it is a priority to ensure a level of continuity of care in primary care. Improvements in care coordination activities are necessary to avoid delays in referrals and diagnosis.

## Figures and Tables

**Figure 1 ijerph-18-11992-f001:**
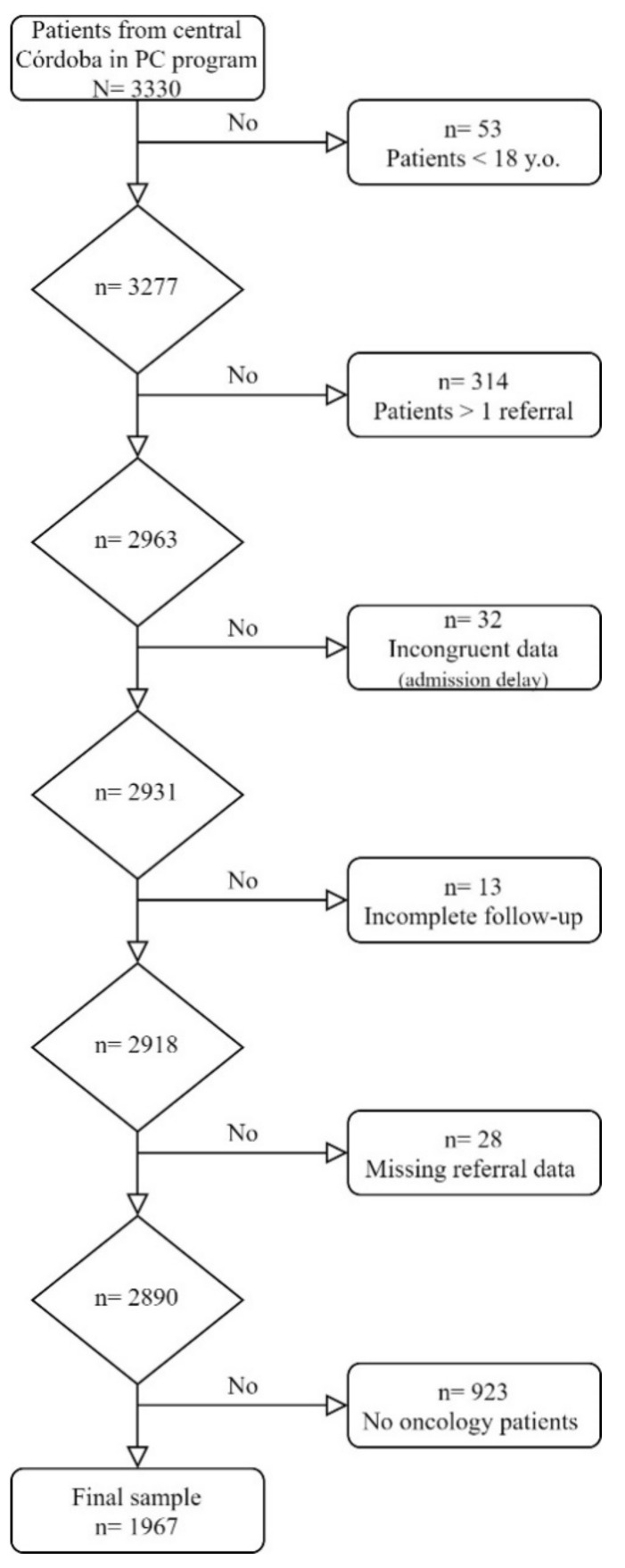
Flow chart of the sample.

**Figure 2 ijerph-18-11992-f002:**
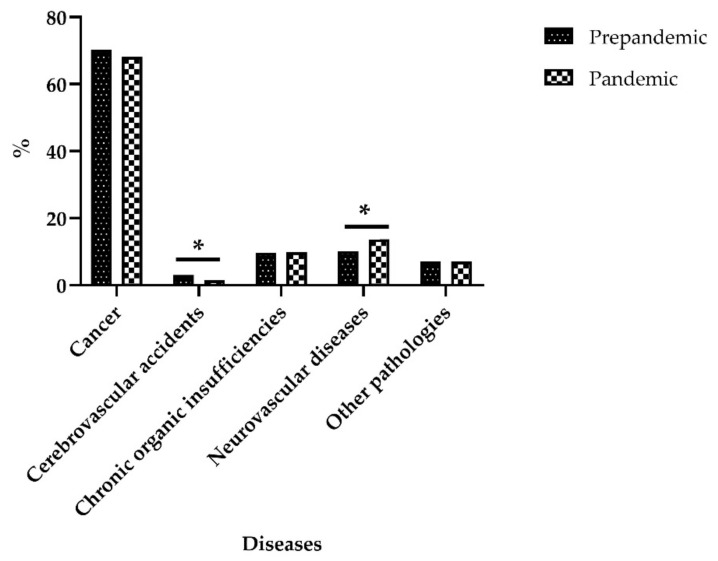
Distribution of diseases treated by PC before and during the pandemic * (*p* = 0.008).

**Table 1 ijerph-18-11992-t001:** Parameters analysed.

**Patients’ socio-demographic data**	Gender	Male/Female
	Age	Years
	Main caregiver	FDR ^1^; SDR ^2^; other relatives; professionals
**Clinical situation of patient**	Cancer	Type of oncological process
	Metastasis	Presence or absence
	Complexity [20]	Non-complex situation; complex; highly complex discharge
	Reason for leaving program	Discharge; death
	Karnofsky Performance Status scale (KPS)	Ability to perform routine tasks (0–100)
	Eastern Cooperative Oncology Group scale (ECOG)	Cancer patient’s quality of life (0–5)
**Health care provided**	Referred from	Primary care; outpatient care; inpatient care; emergency care and other
	CMU ^3^ of origin	Type of CMU
	Derivation priority	Normal; Urgent; Preferential
	Place of death	Home; general hospital; PC hospital; emergency ward; others
	Total delay	Time from patient referral to PC to inclusion in program (days)
	Length of stay in program (patient’s survival)	Time from patient being attended to patient leaving program (days)
**Degree of patient’s knowledge and preferences**	Advance vital directives	Document registered by patient (yes/no)
	Knowledge of patient and family	Degree of patient’s knowledge of real situation (not informed; partial; full knowledge; not applicable)
	Assessment of patient and family	Degree of assessment and understanding of patient’s situation (not informed; partial; full knowledge; not applicable)

^1^ FDR: first degree relative. ^2^ SDR: second degree relative. ^3^ CMU: clinical management unit.

**Table 2 ijerph-18-11992-t002:** Scales analysed.

Scale	Points	Description
**Karnofsky Performance Status scale ^1^** **(KPS)** [21]	0	Dead
	10	Moribund
	20	Completely bedridden, very sick, hospital admission necessary; active support treatment necessary
	30	Severely disabled; hospital admission indicated, and active supportive treatment given
	40	Disabled, requires special care and assistance. Bedridden for over half the day.
	50	Requires considerable assistance and frequent medical. Bedridden for less than half the day.
	60	Requires occasional assistance but is able to care for most personal needs
	70	Cares for self; unable to carry on normal activity or do active work
	80	Able to perform normal activity with effort; some signs and symptoms of disease
	90	Able to carry on normal activity; minor signs and symptoms of disease
	100	Normal, no complaints, no evidence of disease
**Eastern Cooperative** **Oncology Group scale ^2^** **(ECOG)** [22]	0	Completely asymptomatic, fully able to do work and everyday activities
	1	Shows symptoms that do not prevent them from doing their work or everyday activities
	2	Unable to carry out any work activities, with symptoms which force them to stay in bed for several hours a day
	3	Confined to bed or chair for more than half the day due to the existence of symptoms
	4	Totally confined to bed or chair all day and needing help with all everyday activities
	5	Dying or will die within hours

^1^ KPS: Karnofsky Performance Status scale. ^2^ ECOG: Eastern Cooperative Oncology Group scale.

**Table 3 ijerph-18-11992-t003:** Sociodemographic data of the patients before and during the pandemic.

	Pre-Pandemic	Pandemic	*p*-Value
	*n* = 1219 (%)	*n* = 748 (%)	
Gender			0.027
Males	743 (61%)	418 (55.8%)	
Females	476 (39%)	330 (44.2%)	
Age (years)			0.574
Median [IQR] ^1^	75.69 [64.98–83.73]	75.61 [65.25–84.67]	
Caregivers			
FDR ^2^	**1020 (83.7%)**	**657 (88%)**	0.008
SDR ^3^	57 (4.7%)	43 (5.7%)	
OR ^4^	**104 (8.5%)**	**37 (5%)**	
Professionals	**38 (3.1%)**	**11 (1.3%)**	

^1^ IQR: interquartile range, ^2^ FDR: first degree relative, ^3^ SDR: second degree relative, ^4^ OR: other relative. Bold: the bold marks statistically significant differences between both groups.

**Table 4 ijerph-18-11992-t004:** Data on the patients’ clinical situation.

	Pre-Pandemic	Pandemic	*p*-Value
	*n* = 1219 (%)	*n* = 748 (%)	
Type of cancer			0.242
Lung	270 (22.2%)	136 (18.2%)	
Intracranial	37 (3%)	21 (2.8%)	
Haematological	63 (5.2%)	39 (5.2%)	
Prostate	41 (3.4%)	27 (3.6%)	
Urinary Tract	83 (6.8%)	58 (7.8%)	
Colorectal	207 (17%)	138 (18.5%)	
Breast	78 (6.4%)	52 (6.9%)	
Head-neck	63 (5.2%)	23 (3.1%)	
Genitals	72 (5.9%)	51 (6.8%)	
Oesophagus-stomach	87 (7.1%)	48 (6.4%)	
Liver	49 (4%)	33 (4.4%)	
Pancreas-bile ducts	122 (10%)	78 (10.4%)	
Bone	1 (0.1%)	4 (0.6%)	
Others	45 (3.7%)	40 (5.3%)	
Presence of metastasis			0.156
Yes	469 (38.5%)	313 (41.8%)	
Complexity			0.020
Not complex	800 (65.6%)	459 (61.3%)	
1 complex element	151 (12.4%)	70 (9.3%)	
Several complex elements	33 (2.7%)	19 (2.6%)	
Highly complex	**235 (19.3%)**	**200 (26.8%)**	
KPS ^1^ (points)			0.141
10	22 (1.8%)	21 (2.8%)	
20	61 (5%)	26 (3.5%)	
30	168 (13.8%)	70 (9.3%)	
40	261 (21.4%)	153 (20.4%)	
50	464 (38.1%)	318 (42.5%)	
60	191 (15.7%)	121 (16.2%)	
70	27 (2.2%)	17 (2.3%)	
80	11 (0.9%)	17 (2.3%)	
90	9 (0.7%)	1 (0.2%)	
100	5 (0.4%)	4 (0.5%)	
ECOG ^2^ (points)			<0.001
0	12 (1%)	0 (0%)	
1	40 (3.3%)	13 (1.7%)	
2	**631 (51.8%)**	**186 (24.8%)**	
3	**333 (27.2%)**	**389 (52.1%)**	
4	**191 (15.7%)**	**153 (20.5%)**	
5	12 (1%)	7 (0.9%)	
Reason for ending program			<0.001
Discharge	7 (0.6%)	22 (2.9%)	
Death	**1212 (99.4%)**	**726 (97.1%)**	

^1^ KPS: Karnofsky Performance Status scale: a lower score indicates a worse patient survival and quality of life. ^2^ ECOG: Eastern Cooperative Oncology Group scale: a higher score indicates a poorer performance status. Bold: the bold marks statistically significant differences between both groups.

**Table 5 ijerph-18-11992-t005:** Data on the healthcare provided.

	Pre-Pandemic	Pandemic	*p*-Value
	*n* = 1219 (%)	*n* = 748 (%)	
Referral from			<0.001
Primary care	**201 (16.5%)**	**80 (10.7%)**	
Outpatient care	539 (44.2%)	312 (41.7%)	
Inpatient care	**474 (38.9%)**	**348 (46.5%)**	
Emergency and other	5 (0.4%)	8 (1.1%)	
CMU ^1^			0.326
Cardiology	5 (0.4%)	1 (0.2%)	
Plastic surgery	4 (0.2%)	3 (0.6%)	
General surgery	44 (3.6%)	24 (3.2%)	
Thoracic surgery	0 (0%)	1 (0.2%)	
Digestive	126 (10.3%)	74 (9.9%)	
Pain unit	0 (0%)	1 (0.2%)	
Gynaecology	6 (0.5%)	13 (1.7%)	
Home hospital treatment	0 (0%)	1 (0.2%)	
Haematology	51 (4.2%)	31 (4.1%)	
Infectious diseases	2 (0.2%)	1 (0.2%)	
Maxillofacial surgery	6 (0.5%)	2 (0.3%)	
Internal medicine	121 (9.9%)	79 (10.7%)	
Nephrology	2 (0.2%)	1 (0.2%)	
Neurosurgery	13 (1.1%)	8 (1.1%)	
Pneumology	67 (5.5%)	26 (3.5%)	
Neurology	6 (0.5%)	4 (0.6%)	
Oncology	701 (57.5%)	427 (57.2%)	
ENT ^2^	4 (0.3%)	5 (0.9%)	
Oncology RT	23 (1.9%)	17 (2.3%)	
Interventional X-ray ^3^	0 (0%)	1 (0.2%)	
Traumatology	1 (0.1%)	0 (0%)	
UCI	2 (0.2%)	0 (0%)	
Emergencies	2 (0.2%)	0 (0%)	
Urology	32 (2.6%)	23 (3.2%)	
Priority			0.153
Normal (no priority)	1108 (90.9%)	663 (88.7%)	
Urgent (priority 1)	16 (1.3%)	8 (1.1%)	
Preferential (priority 2)	95 (7.8%)	76 (10.2%)	
Delay (days)			0.016
Average [IQR] ^4^	1 [0–0]	1 [0–0]	
Place of death	*n* = 1212	*n* = 726	<0.001
Home	609 (50.3%)	422 (58.1%)	
General hospital	165 (13.6%)	110 (15.2%)	
Palliative hospital	**404 (33.3%)**	**184 (25.3%)**	
Emergencies	17 (1.4%)	8 (1.1%)	
Others	17 (1.4%)	2 (0.3%)	

^1^ CMU: clinical management unit. ^2^ ENT: ear, nose & throat/otolaryngology. ^3^ Interventional X-ray: interventional radiology. ^4^ IQR: interquartile range. Bold: the bold marks statistically significant differences between both groups.

**Table 6 ijerph-18-11992-t006:** ECOG score analysis according to referral location.

		Pre-Pandemic	Pandemic	*p*-Value
Referral from	ECOG ^1^ (points)	*n* = 1219 (%)	*n* = 748 (%)	
Primary care *n* = 281	0	2 (1%)	0 (0%)	<0.001
	1	7 (3.5%)	0 (0%)	
	2	**114 (56.7%)**	**19 (23.8%)**	
	3	**33 (16.4%)**	**43 (53.8%)**	
	4	45 (22.4%)	15 (18.8%)	
	5	0 (0%)	**3 (3.8%)**	
Outpatient care *n* = 851	0	6 (1.1%)	0 (0%)	<0.001
	1	22 (4.1%)	6 (1.9%)	
	2	**298 (55.3%)**	**94 (30.1%)**	
	3	**143 (26.5%)**	**164 (52.6%)**	
	4	68 (12.6%)	45 (14.4%)	
	5	2 (0.4%)	3 (1%)	
Inpatient care *n* = 822	0	4 (0.8%)	0 (0%)	<0.001
	1	10 (2.1%)	7 (2%)	
	2	**215 (45.4%)**	**66 (19%)**	
	3	**157 (33.1%)**	**182 (52.3%)**	
	4	**78 (16.5%)**	**93 (26.7%)**	
	5	**10 (2.1%)**	**0 (0%)**	
Emergency and other *n* = 13	0	0 (0%)	0 (0%)	0.325
	1	1 (20%)	0 (0%)	
	2	4 (80%)	7 (87.5%)	
	3	0 (0%)	0 (0%)	
	4	0 (0%)	0 (0%)	
	5	0 (0%)	1 (12.5%)	

^1^ ECOG: Eastern Cooperative Oncology Group scale: a higher score indicates a poorer performance status. Bold: the bold marks statistically significant differences between both groups.

**Table 7 ijerph-18-11992-t007:** Risk analysis for permanence in the palliative care program.

	HR ^1^	95% CI ^2^	*p*-Value
Pandemic			
Pre-pandemic (reference)			
Pandemic	0.99	0.82–1.20	0.931
Age (decades/years)	0.89	0.84–0.95	<0.001
Gender			
Male (reference)			
Female	0.79	0.67–0.93	0.005
Presence of metastasis			
No (reference)			
Yes	1.48	1.25–1.75	<0.001
KPS ^3^	0.84	0.78–0.91	<0.001
ECOG ^4^	1.16	1.02–1.31	0.024
Referral from			
Primary care (reference)			
Outpatient care	0.74	0.59–0.94	0.012
Inpatient care	1.00	0.78–1.27	0.968
Emergencies and other	0.99	0.23–4.18	0.988
Priority			
Normal (reference)			
Urgent	8.11	2.96–22.18	<0.001
Preferential	1.38	1.06–1.81	0.017

^1^ HR: Hazard Ratio from Cox regression analysis. ^2^ CI: Confidence Interval. ^3^ KPS: Karnofsky Performance Status scale. ^4^ ECOG: Eastern Cooperative Oncology Group scale.

**Table 8 ijerph-18-11992-t008:** Data on patients’ degree of knowledge and preferences.

	Pre-Pandemic	Pandemic	*p*-Value
	*n* = 1219 (%)	*n* = 748 (%)	
ADLW ^1^ registered			1.000
Yes	13 (1.1%)	7 (1%)	
Patient’s knowledge			0.192
Not informed	161 (13.2%)	93 (12.4%)	
Partial knowledge	375 (30.8%)	276 (36.9%)	
Full knowledge	683 (56%)	379 (50.7%)	
Patient’s assessment			<0.001
Not informed	166 (13.6%)	90 (12%)	
Partial knowledge	66 (5.4%)	31 (4.2%)	
Full knowledge	**658 (54%)**	**511 (68.3%)**	
Not known/Not applicable	329 (27%)	116 (15.5%)	
Family’s knowledge			0.351
Not informed	1 (0.1%)	4 (0.6%)	
Partial knowledge	30 (2.5%)	29 (3.9%)	
Full knowledge	1176 (96.5%)	706 (94.4%)	
Not known/Not applicable	11 (0.9%)	8 (1.1%)	
Family’s assessment			0.683
Not informed	76 (6.2%)	54 (7.2%)	
Partial knowledge	29 (2.4%)	10 (1.4%)	
Full knowledge	1100 (90.2%)	673 (90%)	
Not known/Not applicable	15 (1.2%)	10 (1.4%)	

^1^ ADLW: advance directive living will. Bold: the bold marks statistically significant differences between both groups.

## Data Availability

The datasets used and/or analyzed during the current study are available from the corresponding author on reasonable request after approval from all the authors.

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
