# Peer review of "Impact of the COVID-19 Pandemic on Palliative Care in Cancer Patients in Spain"

_ijerph, 2021, doi:10.3390/ijerph182211992_

Round 1
Reviewer 1 Report
Dear authors, first of all, I want to congratulate you on the interestingness of your research topic, since much has been said about the impact of the pandemic on different health issues and the need to expand the coverage of palliative care. Second, the research presents an internal coherence compared to its structure. However, some aspects must improve.
Summary:
Separate each part with a subtitle, and it is not understood where the results start, for example. In addition, the methodology is not specified, making it challenging to evaluate the internal coherence of objectives, results, and conclusions.
Introduction:
Updated evidence is used, and the main actors on the subject are used; the definition of palliative care for contextualization remains pending.
Methodology:
The instrument used, where it comes from, or its validation is not made explicit. Or if it is more than one for the different types of data requested. It should be specified how ethical considerations were safeguarded in data collection and analysis. Not just the approval of the ethics committee.
Results:
A large amount of anonymous data is delivered. For example, table 4 talks about scales that are not named in methodology.
Discussions:
Well used subtitles in the text and updated references. Conclusions: It remains to be explained the study's contribution to public health, and not only to mention that what they thought was not shown.
References:
85% correspond to the last five years and incorporate primary studies and definitions of relevant actors.
Author Response
We thank the reviewer 1 for his/her comments. We feel that in addressing them the paper has been improved.
- Summary: Separate each part with a subtitle, and it is not understood where the results start, for example. In addition, the methodology is not specified, making it challenging to evaluate the internal coherence of objectives, results, and conclusions:
Subtitles and methodology have been added and the final word count has been adapted (lines 15-28).
- Introduction: Updated evidence is used, and the main actors on the subject are used; the definition of palliative care for contextualization remains pending:
The WHO definition of PC from reference number 3 has been included (lines 37-40).
- Methodology: The instrument used, where it comes from, or its validation is not made explicit. Or if it is more than one for the different types of data requested:
The study design is reflected in the first paragraph of item 2.1 of the Material and Methods section: “The study was a retrospective observational cohort study, which analysed the parameters of healthcare provided to patients at the CMU for PC in Córdoba during the COVID-19 pandemic in 2020 and 2021 compared to previous years” (lines 88-90). In addition, item 2.2. explains that the sample came from the CMU for PC internal management database and included all the patients from central Córdoba who had been treated in the period from January 2018 to May 2021 (lines 100-102). The statistical methodology used is described in section 2.3 (lines 125-135). No other instruments were used for the study.
- It should be specified how ethical considerations were safeguarded in data collection and analysis. Not just the approval of the ethics committee:
Item 2.4 Ethical Statements has been added with the requested data (lines 137-142).
- Results: A large amount of anonymous data is delivered. For example, table 4 talks about scales that are not named in methodology:
Table 4 (now Table 2) has been moved to the Material and Methods section. It explains the equivalence of each of the scores of the ECOG and KPS scales analyzed in our study. All the variables studied in our work (including the ECOG and KPS scales) are summarized in Table 1.
- Conclusions: It remains to be explained the study's contribution to public health, and not only to mention that what they thought was not shown:
At the end of the Conclusions section, we have added a paragraph on the implications of the study (lines 537-544).
Reviewer 2 Report
Thank you for the opportunity to review this interesting paper.
Title, abstract, theoretical background and methods are written very well. It is important to detect potential changes in palliative care due to COVID-19 restrictions.
My main concern is that all analyses are based on univariate comparison of parameters so that it is hard to draw valid conclusions. Besides, a lot of potentially dependent variables have been tested in parallel without adjustment of p-values. For example, there had been an increase in referrals from impatient care (Table 5) instead of primary and outpatient care, which might be the reason for the increase in highly complex cases (Table 3) and more severe conditions of the patients (increase in ECOG level 3 and 4).
Some smaller points for the further parts of the paper:
Results:
- In line 108 ff. (methods) it is described which patients have been excluded from the study, and in the limitation part of the discussion you write that the number of excluded cases had been “considerable”. It would be interesting how many patients were excluded exactly, by giving percentages and/ or a flow chart. Patients who did not complete the follow-up and thus have been excluded might have been in a worse medical condition. A non-responder analysis might help to see whether the final sample was representative in terms of diagnoses, age, sex or stage.
- Analysis of diagnoses, l. 127 ff.: If the difference in percentages of cancer patients between pre-pandemic and pandemic was not statistically significant, it should not be labelled as a “slight decrease” but we would assume that the difference is at random (same accounts for the discussion, l. 219 f.).
- Figure 1: Was the p-value exactly the same (0.008) for the two statistically significant differences?
- Table 2/ l. 137 ff.: The overall p-value is a general indicator of group differences. The conclusion from the overall p-value for caregivers that the difference between FDR (4.3% difference) is statistically significant is very likely but we do not know whether the difference for SDR is also statistically significant. It would be an option to create dummy variables for each level of non-dichotomous variables and to indicate individual p-values per level of the variable (also in the following tables).
- 151 ff. and l. 167 ff.: See above, if differences were not statistically significant, statements like “increased slightly” are imprecise.
- Table 3: For KPS and ECOG it would be helpful to indicate the direction of the scaling (“higher values mean better performance”). I think Table 4 should be shifted to the methods section.
- Table 5: How can the difference in delay be statistically significant if it is both the same value?
- Risk and survival analysis (l. 178 ff.): The data is hard to interpret as a shorter stay in the PC program might be due to shorter survival (negative) but also due to an improvement of the condition so that the patient can be cared for at home (positive).
- 185 ff.: Being older and female might indicate that there is no partner to care for the patient?
Discussion:
- The discussion opens up new topics that were not part of the analysis and thus might be shortened substantially and more linked to the results. For example, incidence of cancer is less relevant in the discussion as survival rates (and thus potential need for PC) are very different for the cancer types. Besides, there was no difference in cancer types between pre-pandemic and pandemic in the study anyway.
- Age of patients in PC is also not comparable to age of cancer patients in general.
Author Response
We thank the reviewer 2 for his/her comments. We feel that in addressing them the paper has been improved.
- My main concern is that all analyses are based on univariate comparison of parameters so that it is hard to draw valid conclusions. Besides, a lot of potentially dependent variables have been tested in parallel without adjustment of p-values. For example, there had been an increase in referrals from impatient care (Table 5) instead of primary and outpatient care, which might be the reason for the increase in highly complex cases (Table 3) and more severe conditions of the patients (increase in ECOG level 3 and 4):
The data presented in Tables 2, 3 and 5 (now 3, 4 and 5) are variables collected on admission (except for the place of death) to the palliative care program that attempt to draw conclusions on the differences before and during the COVID19 pandemic.
As can be seen in the results, there are significant differences in several study variables at admission. These variables are process variables. However, the outcome variable that is a priority in a palliative care program is survival in palliative care, a variable for which a multivariate analysis with key variables has been carried out and which allows us to evaluate the differences in the outcome variables.
Although some results of the bivariate analysis suggest an increase in ECOG in patients on admission, which could lead to lower survival, this was not reflected in the multivariate analysis of survival adjusted for the variables described by the reviewer. Nevertheless, we have performed a bivariate analysis between the ECOG variable and the referral location (chi-square test) (new Table 6 and lines 276-278). It showed that the values of ECOG scale 3 (and in some cases 4 and 5) increased significantly, while values 2 decreased. Except for referrals from the emergency department. Corrected typified residues analysis was used to assess the differences between the categories of a variable (absolute values > 1.96) (explanation added in lines 130-131 in 2.3.Data analysis section). So, an adjusted residual that is more than 1.96 indicates that the number of cases in that cell is significantly larger (or significantly lower, according to ±1.96) than would be expected if the null hypothesis were true, with a significance level of 0.05. I attach an interesting link on the type of analysis: https://www.ibm.com/support/pages/interpreting-adjusted-residuals-crosstabs-cell-statistics (modified date: 16 april 2020). For clarification, we have bolded all significant values (above 1.96; absolute value) in Table 6 (and the rest: 3, 4, 5 and 8).
On the other hand with respect to other variables, such as complexity, as indicated in the Andalusian Palliative Care Plan document on complexity: https://www.redpal.es/wp-content/uploads/2018/12/IDC-Pal-2014-Complejidad.pdf , what is not IDC-Pal?
- It is NOT a needs assessment tool.
- It is NOT a prognostic tool for survival, nor does it determine terminal status.
Moreover, the assessment of complexity is highly influenced by the social context, which is highly influenced by the constraints of the pandemic. Therefore, it has not been included in the multivariate analysis of patient survival in the program.
Results:
- In line 108 ff. (methods) it is described which patients have been excluded from the study, and in the limitation part of the discussion you write that the number of excluded cases had been “considerable”. It would be interesting how many patients were excluded exactly, by giving percentages and/ or a flow chart. Patients who did not complete the follow-up and thus have been excluded might have been in a worse medical condition. A non-responder analysis might help to see whether the final sample was representative in terms of diagnoses, age, sex or stage:
A flow chart (new Figure 1) has been added to the Material and Methods section.
- Analysis of diagnoses, l. 127 ff.: If the difference in percentages of cancer patients between pre-pandemic and pandemic was not statistically significant, it should not be labelled as a “slight decrease” but we would assume that the difference is at random (same accounts for the discussion, l. 219 f.):
Both paragraphs have been modified in the Results (lines 148-152) and Discussion (line 326) sections, which is also in accordance with what is referred to in the lines below (lines 336-337).
- Figure 1: Was the p-value exactly the same (0.008) for the two statistically significant differences?
For the variable "Diseases" (5 categories), a comparison was made before and during the pandemic between its categories, using a chi-square test in an h x k table, in this case 5x2 (see Material and Methods section), so that a single p-value was obtained in the test. Corrected typified residues were used to indicate differences in the categories (absolute values > 1.96). In this case it was neurovascular diseases (2.0) and cerebrovascular accidents (-2.0), marked with * in Figure 1 (now Figure 2), that presented the significant results. Therefore, in accordance with the reviewer's indications, we have corrected the previous paragraph with respect to the percentages of patients with cancer, which indeed showed no variation.
- Table 2/ l. 137 ff.: The overall p-value is a general indicator of group differences. The conclusion from the overall p-value for caregivers that the difference between FDR (4.3% difference) is statistically significant is very likely but we do not know whether the difference for SDR is also statistically significant. It would be an option to create dummy variables for each level of non-dichotomous variables and to indicate individual p-values per level of the variable (also in the following tables):
As explained in the previous paragraph, the chi-square test was performed for the variable "caregiver", and the result obtained was a p=0.008. The statistically significant differences for each category were obtained using corrected typified residues, with the following values: "first degree relatives": 2.1; "other relatives": -2.0 and "professionals": -2.0. (For clarification, we have bolded all significant values (above 1.96; absolute value) in Tables 3, 4, 5, 6 and 8).
In any case, we have performed the calculations following the reviewer's indications (using dummies) and the results are consistent with the differences shown by the residuals:
|
|
p value |
|
FDR vs non-FDR |
0.011 |
|
SDR vs non-SDR |
0.29 |
|
OR vs non-OR |
0.027 |
|
Professionals vs non-professionals |
0.022 |
- 151 ff. and l. 167 ff.: See above, if differences were not statistically significant, statements like “increased slightly” are imprecise:
The requested changes have been made to the KPS scale (line 172); the priority (lines 228-229); and the advance directive living will (lines 299). We have verified that the Discussion section was in accordance with the Results section in these parameters.
- Table 3: For KPS and ECOG it would be helpful to indicate the direction of the scaling (“higher values mean better performance”). I think Table 4 should be shifted to the methods section:
Indications on the values of the ECOG and KPS scales have been added as a footnote in Table 4 (formerly Table 3). And Table 2 (formerly Table 4) has been included in the Material and Methods section.
- Table 5: How can the difference in delay be statistically significant if it is both the same value? :
The reason is that when performing the nonparametric Wilcoxon test (lines 132-133), it is evident that the distribution of the data is different. Although the IQRs are the same in both periods, the total delay is very small (75th percentile). But the p result does not imply significance from a clinical point of view, but because of the distribution of the sample (lines 231-232). In fact, in the paragraphs of the Discussion section where we refer to the delay, we point out that there has been no change in the delay (lines 437-438). Below skewness and kurtosis values are displayed:
|
|
Skewness |
Kurtosis |
|
Pre-pandemic |
5.9 |
4 |
|
Pandemic |
3.9 |
2 |
- Risk and survival analysis (l. 178 ff.): The data is hard to interpret as a shorter stay in the PC program might be due to shorter survival (negative) but also due to an improvement of the condition so that the patient can be cared for at home (positive):
Patients were not withdrawn at the end of follow-up. In the initial part of the paper, factors at admission to the palliative care program are analyzed, so that all those admitted to the program are included. In the 2nd part of the article, their survival in the program is explored in a multivariate adjusted manner. Patients who have not reached the event under study (death) at the time of data cut-off are analyzed as censored data as is done in survival analyses.
- 185 ff.: Being older and female might indicate that there is no partner to care for the patient? :
Although the reviewer's hypothesis may be correct, we have not found any reference to support it. The possible explanation for this phenomenon is given in the Discussion (lines 458-472): “As in previous studies, age and gender were considered determinants of patient survival. Older cancer patients usually have a worse prognosis [53], especially in the first months after diagnosis [54]. However, there is little evidence to show the direct and indirect impact of the pandemic on the survival of elderly patients [55,56]. Our results showed survival and shorter stays in PC in younger patients, which could have been due to their more severe clinical situation when admitted to the program. According to Pendyala et al., this evidence could reflect the fact that elderly patients took extra precautions through fear of contagion by COVID-19, which resulted in a higher survival rate. In this study, a positive correlation was shown between age and the frequency of changes in patient care during the pandemic [57]. On the other hand, another factor could be the socio-sanitary conditions resulting from the pandemic and the greater, more aggressive therapeutic treatment given to younger patients. As regards gender, women had a higher survival rate than men, possibly due to the fact that the most frequent cancers in the study (lung, colorectal and pancreas) accounted for 48.3% of the total and presented a worse prognosis in male patients [54].”
Discussion:
- The discussion opens up new topics that were not part of the analysis and thus might be shortened substantially and more linked to the results. For example, incidence of cancer is less relevant in the discussion as survival rates (and thus potential need for PC) are very different for the cancer types. Besides, there was no difference in cancer types between pre-pandemic and pandemic in the study anyway:
We have deleted the first paragraph of the Discussion section regarding cancer incidence and the first paragraph of incidence according to type of cancer in item 4.2.
- Age of patients in PC is also not comparable to age of cancer patients in general:
In item 4.1 on sociodemographic data, we have eliminated the paragraph on age in cancer patients in a generic form.
Reviewer 3 Report
This manuscript faces the impact of Covid-19 on the palliative care of cancer patients in Spain. I suggest some changes that are needed before acceptance of this manuscript, which you can find in the attached file.
Of note, I ask you to restructure the DISCUSSIONS. In fact, the introduction of Discussions leads off the track, as this manuscript indeed regards only cancer patients. You can delete lines 204-213. Then, I suggest starting with a summary of results (the main result is the decrease in cancer patients referring to PC and the second main result is the healthcare provided (lines 309-357) that explains why there was a decrease in this referral due to COVD-19. Then, you can go on by explaining demographic and clinical characteristics and results of survival analysis (lines 358-371).
In the Conclusions, you should indicate the Meaning of the study (implications)
Author Response
We thank the reviewer 3 for his/her comments. We feel that in addressing them the paper has been improved.
- Of note, I ask you to restructure the DISCUSSIONS. In fact, the introduction of Discussions leads off the track, as this manuscript indeed regards only cancer patients. You can delete lines 204-213:
The indicated paragraph has been deleted.
- Then, I suggest starting with a summary of results (the main result is the decrease in cancer patients referring to PC and the second main result is the healthcare provided (lines 309-357) that explains why there was a decrease in this referral due to COVD-19. Then, you can go on by explaining demographic and clinical characteristics and results of survival analysis (lines 358-371):
We have removed some ancillary data and added a short summary of the most relevant results from the Healthcare provided section (4.3) (lines 338-344), respecting the subsections of the Discussion. The order of the subsections of the Discussion is based on the way in which the Results have been presented in the previous section.
- In the Conclusions, you should indicate the Meaning of the study (implications):
In addition to what is stated in the Conclusions, in a paragraph at the end we have added the implications and importance of the study (lines 537-544).
Round 2
Reviewer 2 Report
The authors considered all comments in an appropriate way and the paper has substantially improved after revision.
I just have one remaining question regarding the numbers of excluded patients. Thank you for adding the flow chart. However, I can't find those with missing follow-up ("Those patients who did not complete the follow-up from admission to the PC program until their death or the end of study were also excluded.") in the flow chart. Vice versa, those with more than one referral are shown in the flow chart, but not referred to in the text.
For a better understanding, I would suggest to add the relevant numbers in the text, e.g. "...patients under 18 (n = 53)" and to keep the same order of exclusions in the description and in the flow chart.
Author Response
We thank the Reviewer 2 for his/her comments.
- I just have one remaining question regarding the numbers of excluded patients. Thank you for adding the flow chart. However, I can't find those with missing follow-up ("Those patients who did not complete the follow-up from admission to the PC program until their death or the end of study were also excluded.") in the flow chart. Vice versa, those with more than one referral are shown in the flow chart, but not referred to in the text.
We have completed the text and created a new flowchart to match the information contained in both. When we made the flowchart, we only annotated "admission delay" within the rectangle referred to as "incongruent data" due to lack of space. We have broken down the “incongruent data” (referred only to "admission delay": more than 16 days or even negative data) and those who did not complete the follow-up until their death or the end of study.
- For a better understanding, I would suggest adding the relevant numbers in the text, e.g., "...patients under 18 (n = 53)" and to keep the same order of exclusions in the description and in the flow chart.):
The paragraph of 2.2 Sample and procedure section (lines 107-112) has been amended according to the reviewer's recommendations.
Reviewer 3 Report
The paper has substantially improved. However, the authors should change their manuscript's title into: "Impact of the COVID-19 pandemic on palliative care in cancer 2 patients in Spain"
Moreover, in the Data analysis paragraph, they should define the event (patient's survival or permanence in PC) and censored observations (I think either drop-out from PC or death). Furthermore, they must describe methods used to assess the proportional hazards assumption
Author Response
We thank the Reviewer 3 form his/her comments.
- The authors should change their manuscript's title into: "Impact of the COVID-19 pandemic on palliative care in cancer 2 patients in Spain"
The title has been changed according to the reviewer´s recommendation.
- Moreover, in the Data analysis paragraph, they should define the event (patient's survival or permanence in PC) and censored observations (I think either drop-out from PC or death).
Patient's permanence in the PC program is listed in Table 1 (Healthcare provided section) as a variable. We have added the expression "patient's survival" for clarity.
Regarding the observations, the second paragraph of the 2.2. Sample and procedure section has been improved and includes the cases that were excluded and the different reasons (lines 107-112). In addition, it has been represented in Figure 1 (also modified).
The number of deceased patients is shown in Table 4.
- Furthermore, they must describe methods used to assess the proportional hazards assumption:
Survival analysis was performed using Cox Proportional Hazard methods. It is included in the section 2.3 Data analysis section (line 131) and it has been added as a footnote of table 7.